# Oxidative DNA Damage in the Pathophysiology of Spinal Cord Injury: Seems Obvious, but Where Is the Evidence?

**DOI:** 10.3390/antiox11091728

**Published:** 2022-08-31

**Authors:** Elle E. M. Scheijen, Sven Hendrix, David M. Wilson

**Affiliations:** 1Neurosciences, Biomedical Research Institute, Hasselt University, Agoralaan Building C, 3590 Diepenbeek, Belgium; 2Institute for Translational Medicine, Medical School Hamburg, Germany, Am Kaiserkai 1, 20457 Hamburg, Germany

**Keywords:** spinal cord injury, reactive oxygen species, oxidative stress, antioxidants, oxidative DNA damage, DNA damage response, DNA repair

## Abstract

Oxidative stress occurs at various phases of spinal cord injury (SCI), promoting detrimental processes such as free radical injury of proteins, nucleic acids, lipids, cytoskeleton, and organelles. Oxidative DNA damage is likely a major contributor to the pathogenesis of SCI, as a damaged genome cannot be simply turned over to avert detrimental molecular and cellular outcomes, most notably cell death. Surprisingly, the evidence to support this hypothesis is limited. There is some evidence that oxidative DNA damage is increased following SCI, mainly using comet assays and immunohistochemistry. However, there is great variability in the timing and magnitude of its appearance, likely due to differences in experimental models, measurement techniques, and the rigor of the approach. Evidence indicates that 8-oxodG is most abundant at 1 and 7 days post-injury (dpi), while DNA strand breaks peak at 7 and 28 dpi. The DNA damage response seems to be characterized by upregulation of PCNA and PARP1 but downregulation of APEX1. Significant improvements in the analysis of oxidative DNA damage and repair after SCI, including single-cell analysis at time points representative for each phase post-injury using new methodologies and better reporting, will uncover the role of DNA damage and repair in SCI.

## 1. Spinal Cord Injury

Spinal cord injury (SCI) is a severely disabling neurological disorder of the central nervous system (CNS) that impacts the lives of up to a half-million people worldwide every year [1]. After stroke, SCI is the second leading cause of paralysis in the USA [2], and young adults are especially at risk due to their high-risk behavior [3,4]. SCI is defined as damage to the spinal cord that arises from trauma, e.g., accidents or assault, or from non-traumatic disease or degeneration, e.g., tuberculosis or tumors [4,5]. Disruption of the spinal cord causes a loss of neural circuitry between the cortex and the periphery, leading to decreased motor (including paralysis), sensory, and autonomic function below the injury site. Additionally, the psychological and social state of people with SCI is often poor, leading to depression, anxiety, low quality of life, and social isolation [4,6,7]. All these factors contribute to an estimated therapy cost of over 70,000 USD per patient per year [8], presenting an enormous burden, physically, emotionally, and economically, on patients, their families, and healthcare systems.

The pathophysiology of SCI can be generally divided into two phases: the primary injury event and a secondary exacerbation of the injury site. The initial primary acute trauma destroys the integrity of spinal cord neural circuits and is associated with damage to the local blood vessel architecture and operations. In particular: (i) disruption of the local vasculature causes hemorrhaging, (ii) consequent clotting and edema increase the pressure on the neural and glial circuitry since the spinal cord is positioned within a confined space, and (iii) ischemia results in oxygen and glucose deprivation, suppressing regeneration [9]. The secondary injury phase expands the damaged neural tissue beyond the original borders of the acute injury and, by doing so, worsens the functional decline. Several biochemical events have been proposed to be central to this secondary phase: (i) spinal shock characterized by paralysis, sensory deficits, and absence of reflexes; (ii) vascular dysfunction continuing from the acute phase; (iii) membrane and ionic dysregulation stemming from the hyperpermeability of the cell membrane; (iv) neurotransmitter toxicity due to excessive glutamate release; (v) oxidative stress causing lipid peroxidation and free radical injury of proteins, nucleic acids, cytoskeleton, and organelles; (vi) neuroinflammation as a result of a compromised blood–brain barrier that allows massive infiltration of peripheral immune cells, intensifying the oxidative stress; (vii) glial scar formation by astrocytes that forms a physical and biochemical barrier that blocks axonal regrowth; (viii) activation of Nogo receptors that collapse growth cones and stop neurite elongation [9].

Reactive oxygen species (ROS) and associated oxidative damage challenge recovery following SCI in both phases, mainly due to ischemia in the acute phase and excessive free radical production in the secondary phase. Indeed, oxidative stress causes damage to both lipids and proteins post-SCI [10]. Whereas detrimental oxidative damage to lipids, proteins, and RNA can be resolved by degradation and restorative turnover, chemical alteration to the genetic material can lead to persistent adverse molecular and cellular outcomes dictated by the type of modification. For instance, lesions that change DNA’s coding property can promote mutational events that permanently alter the genetic code, potentially dysregulating growth-control genes and initiating carcinogenesis. More complex lesions that block the progress of RNA or DNA polymerases can result in arrested transcription or replication, respectively, leading to senescence or cell death—fates that culminate in degenerative diseases typically associated with aging [11]. The CNS is especially prone to degeneration following persistent DNA damage accumulation, primarily as a result of the post-mitotic character of neuronal cells but also due to the limited DNA repair repertoire associated with non-proliferating cells [12]. The review here will present data that support the role of oxidative DNA damage in the pathogenic process of an SCI, though findings are often contradictory, with broad gaps in our current knowledge.

## 2. Oxidative Stress and Antioxidants following SCI

Under normal physiological conditions, the spinal cord, and the CNS more broadly, produce high concentrations of free radicals due to the metabolic nature of neurons. To meet the intense energy demands, mitochondria work relentlessly to generate ATP via oxidative phosphorylation, with ROS being generated as by-products of respiration. ROS leakage into the intracellular environment becomes more pronounced with age or during specific disease states due to lost mitochondrial integrity or mitochondrial dysfunction [13,14]. Thus, mitochondria are a constant source of ROS within the spinal cord, and since there is also a high amount of transition metals (i.e., copper and manganese) [9], the spinal cord is continuously under threat of oxidative stress, particularly after injury.

Following SCI, excessive ROS formation in and around the injury site leads to an extreme oxidative environment. First, an even greater energy supply is needed within the spinal cord following injury to support repair, increasing mitochondrial ROS production [15]. Second, mitochondria may be damaged and/or become dysfunctional, enhancing the already high ROS formation and leakage into the intracellular milieu [16]. Finally, infiltration of peripheral immune cells into the injured spine, particularly during the secondary phase, increases the oxidative load, as immune cells, e.g., macrophages, use ROS to clean away debris [9].

In early SCI studies in mice, excessive ROS formation, including hydrogen peroxide (H_2_O_2_), superoxide (O_2_^●−^), hydroxyl radical (^●^OH), nitric oxide radical (NO^●^), and peroxynitrite (ONOO^−^), has been shown to occur 1 to 24 h post-injury (hpi) [17,18,19,20,21,22]. Subsequent studies indicate that ROS are present up to at least 10 days post-injury (dpi) [23,24,25]. ROS have been shown to attack several macromolecules within the spinal cord following injury, causing higher levels of RNA/DNA oxidation, protein oxidation, and lipid peroxidation [10,19,20,23,26,27,28,29]. Lipid peroxidation can cause self-propagating membrane damage that can induce cell death in primarily affected cells and nearby healthy cells [9]. Additionally, SCI causes iron metabolism dysfunction [30], leading to iron overload within the spinal cord. Iron is known to cause oxidative damage via the Fenton reaction, which involves hydrogen peroxide. Since iron has a propensity to bind DNA through ionic interactions, there is an increased likelihood of genomic damage during periods of iron overload and elevated ROS [31]. Furthermore, myeloperoxidase activity increases following SCI, indicating ROS production by infiltrating immune cells [23,26]. Other oxidative stress-related proteins are also found prominently within the spinal cord following injury, such as glutathione S-transferase Yb-3 and apolipoprotein A-I precursor peroxiredoxin-6 [32], consistent with a highly oxidative environment.

Considering that ROS are regularly formed under physiological conditions, cells have evolved protective mechanisms to prevent the adverse damaging effects of ROS. The primary front-line defense platform encompasses a battery of scavenging molecules and enzymes that operate to neutralize ROS, collectively called antioxidants. Unfortunately, the spinal cord possesses moderate levels of endogenous antioxidants and antioxidative enzymes [9]. Upon SCI, most antioxidants, i.e., glutathione (GSH), glutathione S-transferase A1 (GST-a1), NAD(P)H:quinone acceptor oxidoreductase 1 (NQO1), and Cu/Zn superoxide dismutase (SOD1), are unexpectedly decreased within the spinal cord, contributing to the excessive oxidative stress [20,23,26,33,34]. Not surprisingly, treatments that increase local antioxidant levels have proven successful in preclinical SCI studies. For example, intraperitoneal (i.p.) injections with gallic acid or Trolox, oral administration of riboflavin, intravenous (i.v.) injections with MCI-186, and lycopene treatment (administration route not defined) all lowered local ROS presence and increased recovery following SCI in rodents [25,34,35,36,37]. Other therapeutic paradigms using antioxidant strategies such as transgenic mice overexpressing SOD1, oxidation resistance 1 (OXR1) gene delivery via liposomal nanoparticles, and spinal cord perfusion with Mn(III) tetrakis(4-benzoic acid) porphyrin (MnTBAP) similarly improved post-SCI parameters [29,36,38,39]. Nuclear factor erythroid 2-related factor 2 (NRF2) is a transcription factor that regulates the expression of genes harboring a promoter antioxidant response element, thereby protecting cells from the consequences of oxidant exposure [40]. Therapeutic activation of NRF2 improved recovery following SCI, while NRF2 knockout mice experience exacerbated damage [33,41]. Altogether, the massive oxidative stress created after an SCI must be dampened to allow proper repair and prevent pathologic neurodegeneration.

## 3. Oxidative DNA Damage Is Understudied after SCI

As noted above, when antioxidants are insufficient or free radical production is excessive, unresolved ROS will react with any neighboring macromolecule, such as lipids, proteins, RNA, and DNA, creating unwanted modifications [42,43]. Whereas detrimental damage to the first three can be resolved by degradation and turnover, chemical alterations to the genetic material can lead to permanent or persistent adverse molecular and cellular outcomes. In many situations where the DNA damage is too severe or DNA repair is insufficient, cells will die. DNA damage-related cell death mechanisms include necrosis, autophagy-mediated, and caspase-dependent or -independent apoptosis [44]. We focus this section on the evidence that shows that modifications to DNA are increased following SCI.

### 3.1. Limited Evidence for Base Lesions

Oxidative DNA damage is a broad term for any DNA lesion caused by ROS attack on the DNA molecule. By far the most famous and best-studied oxidative base lesion is 8-hydroxy-2′deoxyguanosine (8-oxodG), although at least 20 other oxidative base modifications have been identified in a stable form [45]. 8-oxodG is generated explicitly by ROS attack of the C8 position in guanine and, due to its mispairing potential with A, can lead to G:C to T:A transversion mutations. With the existence of several established quantification methods, studies on oxidative DNA damage and SCI have focused mainly on 8-oxodG (summarized in Table 1). Though most studies used 8-oxodG as a readout for establishing the efficacy of a tested therapy following SCI, information can be extracted from this work regarding the production of oxidative base damage. All papers agree that 8-oxodG formation is increased following SCI, although the specifics vary considerably [10,34,36,37,46,47,48,49,50].

While studies consistently support that oxidative base damage increases following an SCI, the precise timing of 8-oxodG formation is unclear (see Figure 1A, which shows all papers that have analyzed oxidative base lesions). Significant increases of 8-oxodG have been reported at any time point post-injury, ranging from 3 hpi up to 21 dpi. Leski et al., who were the first to investigate oxidative base damage after SCI, found, using high-performance liquid chromatography (HPLC), that out of the six tested time points ranging from 1 hpi to 2 dpi, only 3 and 5 hpi had significant increases in 8-oxodG [50]. Later, Bao et al. and Martin et al. showed via immunohistochemistry (IHC) that all tested time points between 1 and 10 dpi were significantly increased for the base lesion [36,47]. Other techniques, such as Western blotting (performed by Kotipatruni et al.) and ELISA (performed by Sakarcan et al.), confirmed the constitutive increase in 8-oxodG following injury and expanded the timeframe to 21 dpi [34,49]. Although one should be careful about comparing data between research groups and different experimental assays, it is apparent that there is currently no consensus regarding the magnitude of fold increase at the different time points post-injury (see Figure 1B, which shows only papers that have quantified oxidative base lesions). Nevertheless, the overall picture is consistent with an oxidative environment.

In addition to the quantitative data above, experiments lacking quantitative assessments or proper comparative controls were also conducted to evaluate 8-oxodG levels in SCI tissue. Using IHC approaches, Sakurai et al. and Takahashi et al. found no signal increase versus sham at 8 hpi, while at 1 and 2 dpi, an increase was visible although not statistically validated [37,46]. Xu et al. observed a non-quantified increase of 8-oxodG at 1, 6, 12, and 24 hpi [29]. Finally, two studies quantified 8-oxodG IHC but did not compare the results to a Sham group but instead to a treatment group. King et al. and Huang et al. observed the presence of 8-oxodG at 7 dpi and 3 and 7 dpi, respectively [10,48]. Thus, despite the different experimental techniques used by different research groups, the data indicate hotspots of 8-oxodG at 1 and 7 dpi. Figure 1A, which summarizes the collective of the studies, indicates that despite most experiments revealing increased 8-oxodG at 1 hpi, 12 hpi, 1 dpi, and 2 dpi, there are also studies showing no difference between SCI and Sham groups, indicating the need for more detailed and comprehensive studies assessing the profile and timing of oxidative base damage following SCI.

#### 3.1.1. Experimental Challenges of Base Lesion Assays

Several notable gaps exist in the current knowledge on oxidative base lesions post-SCI. Generally speaking: (i) no data exist beyond 21 dpi, (ii) the origin of the examined tissue is unknown, (iii) the affected cell type is unknown, (iv) the type of damaged nucleic acid is unknown, (v) antibody-based techniques are mainly used, and (vi) no other oxidative base lesions (apart from 8-oxodG) are assessed. Since no data are available beyond 21 dpi, analysis of the early chronic phase of an SCI has been excluded so far. This period is characterized by immune cell influx and associated ROS formation and is, therefore, a very relevant moment to analyze the effects of oxidative stress on DNA. Additionally, there is often no reporting on the origin of the examined tissue: is it from the lesion site (one would presume) or perilesional, and if so, at what distance from the lesion? Additionally, to date, there has been almost no cell-typing of 8-oxodG+ cells—are these motor neurons, glial cells, or infiltrating immune cells? Likewise, 8-oxodG can exist within nuclear DNA, cytoplasmic RNA, or even mitochondrial nucleic acids, indicating a need for determining the origin of the detected 8-oxodG within the cell. Additionally, the resolution of the exact location of the DNA damage within the genome is very poor. Finally, applying other techniques to quantify 8-oxodG lesions might be worthwhile, such as an adapted comet assay, HPLC, or LCMS/MS. In addition, other oxidative base lesions, e.g., formamidopyrimidine (FaPy) modifications, thymine glycol, or cyclopurines, should be investigated to achieve a complete picture of oxidative DNA damage beyond 8-oxodG. Some existing gaps could have been resolved by better reporting of the experimental design, but other experimental techniques with higher resolution are ultimately needed. Altogether, it seems general knowledge that oxidative DNA damage increases following SCI, yet the evidence and the whole picture remain limited.

### 3.2. Unclear Timing of Strand Breaks

In addition to base lesions (see Section 3.1), oxidative DNA damage encompasses a range of strand breaks that create discontinuity in the phosphodiester backbone of the duplex structure. Such strand breaks are formed both by direct attack of ROS and as intermediates of DNA metabolic events, potentially causing genomic instability and chromosomal aberrations if unresolved. DNA strand breaks have been measured in the context of SCI primarily by the comet assay [49,51,52,53,54,55], but also by poly(ADP) ribose (PAR) IHC [56,57] and γ-H2AX IHC [58]. The comet assay, or the single-cell gel electrophoresis (SCGE) technique, is a method for obtaining a general picture of DNA strand breaks within the genome. In brief, a single-cell suspension is embedded within agarose, and following lysis, the DNA is electrophorized using optimized parameters, producing comet-like genomic structures. The head of the comet represents intact DNA, whereas the comet’s tail contains DNA fragments arising from strand breaks. Parameters deducted from the comet assay are the tail% (tail DNA content), tail length (measured from the center of the head to the end of the tail), and tail moment (tail% × tail length), all of which provide a quantitative measure of the degree of DNA breakage (single and double, and possibly alkaline-sensitive sites) within the single-cell genome. IHC, conversely, is a precise method to analyze strand breaks. In the case of PAR IHC, the signal reflects the polymer produced by poly(ADP) ribose polymerase (PARP) as part of its nick sensor response [59]. γ-H2AX IHC identifies the phosphorylated form of histone H2AX, which is a marker of DNA double-strand breaks (DSBs) that arises via the kinase activity of ataxia telangiectasia mutated (ATM) or ATM-Rad3-related (ATR) [60]. Like 8-oxodG, there is a consensus that strand breaks increase following SCI (Table 2).

As with 8-oxodG, the timing of strand breaks is again unclear (see Figure 2A, which shows all published papers on DNA strand breaks post-SCI). Typically, time points ranging from 5 to 28 dpi have been analyzed, with the earlier investigations being more qualitative in nature. Specifically, Liu et al. established the presence of DNA strand breaks via the comet assay following SCI at 5–10 dpi, although their experiment did not include a comparative Sham group [51]. Martin et al., Genovese et al., and Dagci et al. 2009a reported qualitatively more DNA strand breaks in SCI groups compared to Sham groups at time points ranging from 1 to 28 dpi yet did not include rigorous statistical analyses [52,53,56]. Later studies by Dagci et al. 2009b and Ozgonul et al., which included a Sham group and applied statistics, showed significant increases in strand breaks via the comet assay at 7 and 28 dpi [54,55]. Significant increases were observed by Paterniti et al. using PAR IHC at 1 dpi [57]. In comparison, higher strand break levels were seen by Kotipatruni et al. at 21 dpi via the comet assay, with no significant changes at 1 and 7 dpi [49]. Finally, one study performed by Tuxworth et al. looked explicitly at DSBs using γ-H2AX IHC, establishing the presence of DSBs at 28 dpi, yet without comparison to a Sham group but instead a treatment group [58].

In addition to comet assays and IHC, TUNEL assays and other DNA fragmentation methods (e.g., fragmented DNA separation and diphenylamine reaction) can be considered measurements of DNA strand breaks. DNA gel electrophoresis and TUNEL assays have been performed on SCI tissue since 1999 and are still being performed to date. With such techniques, it has been repeatedly found that DNA fragmentation occurs in the spinal cord following injury at 1 to 7 dpi [17,23,26,48,56,61,62,63,64,65,66,67,68,69,70,71,72]. Though one cannot distinguish in these experiments between strand breaks in viable cells that can be resolved by DNA repair and DNA fragmentation as a consequence of apoptosis, the data add to the evidence of strand breaks occurring post-SCI.

Strand breaks have also been assessed in in vitro cell culture assays using agents that mimic essential aspects of the SCI pathophysiology, such as NO^●^, H_2_O_2_, ONOO^−^, and kainate, on primary motor neurons or spinal cord preparations. Though these assays do not encompass the full complexity of an SCI in vivo and merely mimic important aspects of the SCI lesion, it was found that primary neurons accumulate general strand breaks [73], single-strand breaks (SSBs) [51,74], and DSBs [75] following genotoxic stress. These data indicate, not surprisingly, that oxidizing agents and ROS in and of themselves can cause DNA damage in spinal cord motor neurons.

#### 3.2.1. Experimental Challenges of Strand Break Assays

Studies performed on DNA strand breaks, such as 8-oxodG, have several knowledge gaps, including: (i) conflicting results at specific time points, (ii) unclear magnitude of the fold change, (iii) no data before 1 dpi, (iv) no information about the origin of the examined tissue, (v) no details regarding the affected cell type, (vi) no certainty about the affected nucleic acid, and (vii) only one study on DSBs. When comparing studies, it becomes apparent that contradictory findings are found at 1, 7, and 28 dpi, as Kotipatruni et al. and Martin et al. found no statistical difference between Sham and SCI mice. In contrast, Dagci et al. 2009b, Paterniti et al., and Ozgonul et al. showed significant differences at these time points [49,52,54,55,57]. Apart from the timing itself, the magnitude of fold change at specific time points is also unclear (see Figure 2B, which reports only studies that quantified DNA strand breaks). Especially at 7 dpi, there seems to be no agreement between the different studies on the magnitude of the difference between SCI and Sham groups using the comet assay, as it ranges from 1.7 to 35. In general, data are sparse at the earlier time points, i.e., 3 dpi and earlier, and hyper-acute pathophysiology is completely missing, as there exists no data earlier than 1 dpi. As DNA damage and repair are fast processes, occurring within minutes to hours, these early time points are essential for future studies [76]. Like 8-oxodG, no cell-typing, tissue origin, or cellular location of the DNA strand breaks is reported. Data on DSBs in particular is lacking, as only one paper reports findings on γ-H2AX staining. In general, DNA strand breaks appear after SCI; however, the timing and resolution remain limited.

### 3.3. SCI Causes Systemic DNA-Damaging Effects

While it can be appreciated that an SCI can have local DNA-damaging effects within the spinal cord, evidence also indicates that an SCI can induce systemic genotoxicity. Notably, several researchers have found DNA-damaging effects following SCI in bodily fluids in rats. Using HPLC, Ozgonul et al. found DNA damage biomarkers (i.e., 5-(hydroxymethyl) uracil (5HMU) and 2′-deoxyuridine (2dU)) in the urine at sub-acute time points (7 dpi) and early chronic time points (28 dpi) [55]. In addition, Medalha et al. showed DNA strand breaks using the comet assay in the blood of SCI rats at 1 dpi [77]. These findings indicate that factors produced at the injury site can traverse the disrupted blood–brain barrier and may have systemic effects. Indeed, other studies indicate that increased DNA damage is present within organs outside the spinal cord. Dagci et al. 2009a found DNA strand breaks in the brain and kidney of SCI rats [53]. Consistently, Medalha et al. showed DNA strand breaks in the liver and kidney [77], and Sakarcan et al. similarly reported oxidative DNA damage (i.e., 8-oxodG) in the kidney [34]. The brain is directly connected to the spinal cord, and similar effects as in the spinal cord might be predicted. In contrast, the kidney and liver are distant organs that can unexpectedly experience the detrimental effects of an SCI as well. Knowing that (i) the blood–brain barrier is disrupted after SCI and DNA damage occurs outside of the CNS, and (ii) DNA damage induces cellular senescence and apoptotic cell death, it would be worthwhile to assess for these pathological outcomes both in circulation and at distant organs after injury.

## 4. The DNA Damage Response Is Understudied in SCI

As described above, different DNA lesions arise following SCI, both within and outside the affected spinal cord area. If these lesions remain unresolved, several detrimental outcomes could occur, such as mutagenesis, DNA replication fork collapse, or RNA transcription blockage, leading to cellular outcomes such as transformation, apoptosis, or senescence. Fortunately, organisms have developed several repair mechanisms to resolve DNA damage and restore DNA to its natural state [11,12]. Depending on the type of DNA lesion, a different mechanism is called upon. Oxidative DNA lesions, particularly base modifications and sites of base loss, will mostly be resolved by base excision repair (BER) and, in specific cases, nucleotide excision repair (NER), which deals mainly with helix-distorting adducts, e.g., cyclopurines. SSBs are resolved by specialized pathways of single-strand break repair (SSBR). Homologous recombination (HR) tends to DSBs in mitotic cells, namely during the S phase when a homologous template sister chromatid is present, and non-homologous end-joining (NHEJ) clears double-strand breaks in the absence of a homologous chromosome partner (e.g., in G1) or in non-proliferative cells, e.g., neurons. In addition, there is mismatch repair (MMR) for DNA replication (DNArep) errors, such as base–base mismatches or small insertions/deletions, and the Fanconi anemia (FA) pathway for handling replication-blocking lesions, most notably DNA inter-strand crosslinks.

To know whether DNA repair might play a protective role following SCI, a detailed view of DNA repair gene expression following an SCI is needed. Several studies have revealed a differential expression of DNA repair genes after SCI induction (Table 3, Figure 3). Most of the work was performed by Kotipatruni et al. They found that components of BER, i.e., 8-oxoguanine DNA glycosylase (OGG1), thymine DNA glycosylase (TDG), apurinic/apyrimidinic endodeoxyribonuclease 1 (APEX1), and X-ray repair cross-complementing protein 1 (XRCC1), are more abundant in SCI groups versus Sham, as determined by qPCR, Western blot, and IHC [49]. However, three other independent studies have found APEX1 to be less abundant after SCI using one or two of the same techniques [46,47,54]. Kotipatruni et al. also found that SSBR and HR-related factors, i.e., poly(ADP-Ribose) polymerase 1 (PARP1), breast cancer gene 2 (BRCA2), and ataxia-telangiectasia mutated (ATM), were upregulated following SCI [49]. In agreement, two additional studies have reported that PARP1 is upregulated after injury using qPCR, Western blot, and/or IHC [68,78]. One other key HR protein, breast cancer gene 1 (BRCA1), has likewise been reported to be more highly expressed in microglia of SCI groups using RNA sequencing [79]. Proliferating cell nuclear antigen (PCNA), a central replication factor and an auxiliary protein to several DNA repair pathways, has been reported to be upregulated in three independent studies [80,81,82]. The DNArep protein mini-chromosome maintenance complex component 7 (MCM7), the FA-related Fanconi anemia complementation group D2 (FANCD2), and ataxia-telangiectasia and rad3-related (ATR) were found to be upregulated as well [49,79,81]. Finally, studies with human subjects show that in the cerebrospinal fluid (CSF), DNA repair processes are differentially abundant in SCI patients [83]. Overall, the studies to date indicate a general upregulation of DNA repair factors, implying a need for increased DNA repair capacity to presumably cope with the elevated levels of DNA damage.

### Experimental Challenges of DNA Damage Response Assays

The most researched pathway in SCI is BER, likely since BER has a central role in resolving a wide range of oxidative DNA base and sugar lesions. However, it is essential to investigate other DNA repair pathways in greater depth, particularly NER and NHEJ, as these are highly relevant to non-dividing cells such as neurons [12]. Moreover, only two studies performed a cell type-specific analysis [46,79], whereas the rest analyzed DNA repair in the total spinal cord. This limits the translation of the knowledge gained to therapeutic targets as SCI symptoms primarily reflect the degeneration of neurons. Nonetheless, many different techniques (i.e., RNA sequencing, qPCR, Western blot, and IHC) were used within and between studies, increasing confidence in the findings. In total, evidence shows that following SCI, within the BER pathway, TDG and OGG1 are upregulated, and APEX1 is downregulated. Other well-studied factors are PCNA and PARP1, both upregulated due to SCI, suggesting an increased need for genome surveillance and repair mechanisms following injury induction.

## 5. DNA Repair Is an Attractive Therapeutic Target in SCI

Currently, limited effective treatments are available for SCI. Acute management primarily focuses on securing the airway and surgically removing bone and disk fragments inside the spinal cord [84,85]. Following this initial treatment, corticosteroids and anticoagulants are administered to reduce inflammatory cytokine release and oxidative stress and prevent venous thromboembolic events. These procedures are followed by physical therapy to improve independence in daily living and reintegration into society [86]. Besides the general clinically approved therapies (Figure 4), new state-of-the-art treatments are being developed to stimulate neuroprotection and neuroregeneration or improve function using neuroprosthetics or neurostimulation [87,88,89,90]. Despite these continued efforts to find an effective treatment, current strategies do not cure SCI and induce limited functional recovery, highlighting the need for alternative and complementary approaches to treat SCI patients.

Applying DNA damage profiling and DNA repair gene expression knowledge is essential to developing a combinatorial therapy that targets the removal of oxidative DNA lesions, thereby limiting secondary injury after SCI. For example, as mentioned above, PARP1 is upregulated. This outcome seems logical as DNA strand breaks need to be resolved, and PARP1 is a key initiating factor for SSBR. PARP1 detects strand breaks and instructs for repair via PARylation, i.e., the covalent attachment of ADP-ribose moieties to target proteins using NAD+ as a substrate. NAD+ is essential to all cells as it is central to energy metabolism and the production of ATP. Therefore, NAD+-depletion by overactivation of PARP1 could be detrimental to cell survival. Scott et al. demonstrated that the deleterious effects of oxidative stress in spinal cord neurons can be dampened by inhibiting PARP1 in vivo [74]. Moreover, inhibition of PARP1 in mice using different small molecule inhibitors reduced inflammation and neurodegeneration, leading to improved functional recovery following SCI [56,68,70]. PARP1 inhibition even seems to increase antioxidant levels in the spinal cord of mice [78]. Conversely, one study showed that inhibiting PARylation by PARP1 gene inactivation in mice does not improve functional recovery, suggesting differing effects of enzyme inhibition versus enzyme absence [91]. It is possible that inhibiting PARP1 using small molecule inhibitors is sufficient for counteracting NAD+ depletion while leaving some residual PARP1 functionality for resolving DNA strand breaks. Complete ablation of PARP1 by gene deletion would counter the overconsumption of NAD+. However, it would prevent any PARP1 functionality, likely leading to strand break-induced cell death. Similar to the PARP1 inhibition paradigm, another study investigated the effect of DSB repair (DSBR) attenuation on CNS neurodegeneration and found that small molecule inhibitors can promote axon regeneration after SCI [58]. These studies prove that there might be merit in targeting DNA repair players, in the cases above, suppressing overactivation of the DNA damage response to improve recovery following SCI.

### Experimental Challenges of Therapeutic Studies

PARP1 and DSBR studies aside, there are no studies explicitly assessing therapeutics based on supporting the effectiveness of DNA repair pathways. Following the argumentation discussed above, on one hand, one could imagine a therapy that suppresses DNA repair factors to benefit SCI by preserving a fundamental level of energy supply. On the other hand, a therapy boosting DNA repair might reduce the harmful consequences of the DNA damage inflicted by an SCI, thereby preventing neuronal cell loss. In any case, there are significant gaps in knowledge, such as the comparative protective roles of DNA repair mechanisms after SCI, that provide ample opportunities for the development of DNA repair-targeted therapies for SCI.

## 6. Future Outlook

SCI is a complex disorder that includes multiple pathophysiological processes during primary and secondary phases. In this review, we presented the surprisingly limited evidence for oxidative DNA damage and DNA repair in SCI, mainly in the secondary phase (Figure 5). Following the primary injury, infiltration of activated immune cells together with an increased energy demand causes a significant increase in ROS. Combined with the low availability of antioxidants, these ROS attack many macromolecules. Oxidative base lesions, specifically 8-oxodG, are increased primarily at 1 and 7 dpi, and DNA strand breaks are elevated at 7 and 28 dpi. However, mixed results exist throughout the various studies (as discussed earlier). Apart from local DNA-damaging effects, systemic DNA damage in cells of the blood, urine, kidney, liver, and brain can also be found following SCI. The body responds to SCI, and presumably the associated DNA damage, by mainly upregulating a few BER components to seemingly increase oxidative DNA damage responses. Other critical factors in response to SCI seem to be PCNA (a replication factor with several accessory roles in many DNA repair pathways) and PARP1 (a major strand break response factor), while APEX1 (a central BER player) is downregulated.

### 6.1. Oxidative DNA Damage

In the face of SCI, the presence and magnitude of DNA damage have been studied by different research groups using different SCI models as well as diverse DNA damage assays. This is important as it provides both validation and verification of the findings. As the collective results indicate, it is logical that oxidative DNA damage is increased following SCI, considering the intrinsic high oxidative stress and the already well-characterized protein oxidation and lipid peroxidation [9]. However, as described in detail earlier, there are several limitations regarding the studies that have measured oxidative DNA damage following SCI (see Section 3.1.1 and Section 3.2.1). To overcome some of these limitations, one extensive study assessing a wide variety of DNA damage types (e.g., different oxidative base lesions, apurinic/apyrimidinic sites, SSBs and DSBs, and intra-/inter-strand crosslinks) in different cell types at regular intervals that encompass the main phases of SCI using new, state-of-the-art techniques, would provide a solid basis for the next stage of research. One such technique is genome-wide profiling using next-generation sequencing [92]. It has some preference over more traditional assays as it provides nucleotide resolution, revealing the exact location of the target lesion and paving the way for the identification of preferential hot spots of DNA damage in motor neurons (or other cell types) post-SCI.

### 6.2. DNA Repair

Very few studies have investigated DNA repair mechanisms in the face of SCI, and the studies performed have been limited to examining gene expression changes following SCI. Of all pathways and factors within them, only a handful of the repair components have been assessed. Of the work thus far, PCNA and PARP1 are consistently upregulated, while most studies performed on APEX1 show a downregulation. A relatively straightforward solution to addressing the limited picture thus far would be to carry out single-cell omics analysis for DNA repair gene expression or multiplex IHC on SCI tissue relative to Sham controls. Whatever the approach, all pathways and all factors (or at least key factors) should be included to obtain a complete picture. Additional critical investigations would be to determine which DNA repair pathways are most important to providing neuronal cell protection following SCI, as these systems could be part of future therapeutics to identify at-risk individuals and molecular targets to reduce the expansion of the disease following injury.

### 6.3. DNA Repair-Based Therapies

Currently, no study has been performed to assess DNA repair-based therapies in the context of SCI. However, research assessing PARP1 and DSBR inhibition or knockout shows that dampening the DNA damage response might benefit SCI recovery, presumably by preventing hyperactivation and consequent cell death. When the results of general studies to map the spectrum and robustness of DNA damage and the protective role of the different DNA repair mechanisms after SCI have been completed, therapeutic targets and strategies can be more readily defined. Once identified, it will need to be established whether the priority targets require upregulation or inactivation in vivo to promote neuronal cell survival and a beneficial clinical outcome. Likely, some pathways will require inhibition to support normal cellular homeostasis (see PARP1), while others will need stimulation to improve DNA damage resolution and prevent excessive neuronal degeneration.

### 6.4. Conclusions

To date, investigations on oxidative DNA damage in SCI have been limited regarding DNA lesions’ types, phasing, and cell-typing. In addition, not all DNA repair mechanisms have been studied. Further fundamental research, using new methodologies that span the full spectrum of DNA damage/repair, will be necessary to guide future therapeutic targets and strategies. Currently, we know that oxidative DNA damage is taking place. However, the evidence is surprisingly meager, and many gaps in knowledge of DNA damage and repair must be filled as we move forward.

## Figures and Tables

**Figure 1 antioxidants-11-01728-f001:**
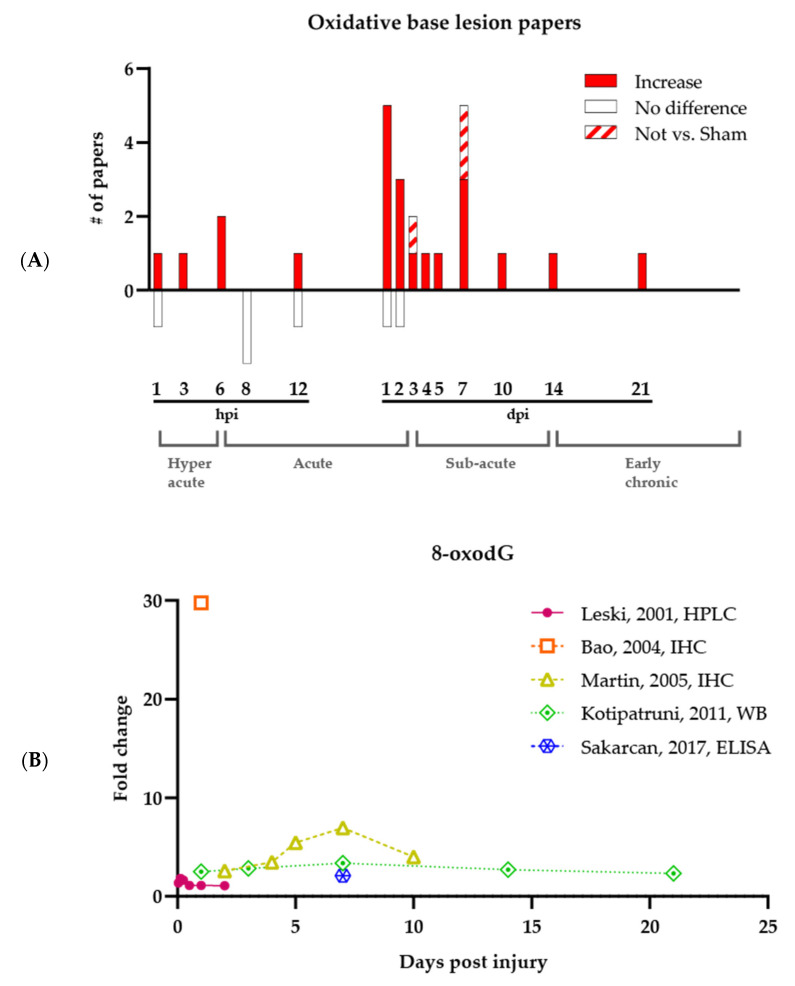
Oxidative DNA damage is increased after SCI. (**A**) Summary of all published papers on oxidative base lesions following SCI. The bars underneath the graph represent papers reporting no difference between SCI and Sham groups. The brackets at the bottom of the figure indicate the different SCI phases. (**B**) Plot of all quantified oxidative base lesion measurements showing the timing and magnitude of 8-oxodG difference between SCI and Sham groups. Calculated fold change of SCI versus Sham per time point is visualized. hpi, hours post-injury; dpi, days post-injury.

**Figure 2 antioxidants-11-01728-f002:**
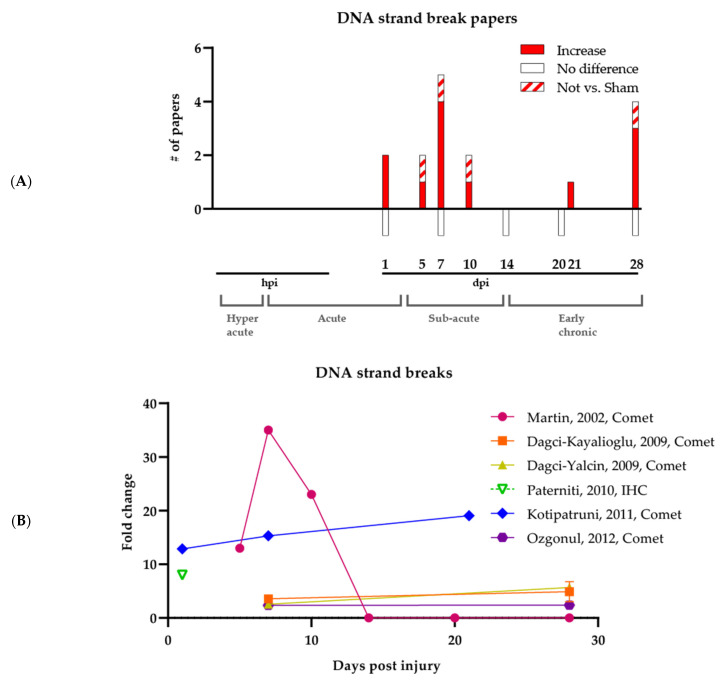
Increase of DNA strand breaks following SCI. (**A**) Summary of all published papers on DNA strand breaks following SCI. The bars underneath the graph represent papers reporting no difference between SCI and Sham groups. The brackets at the bottom of the figure show the different SCI phases. (**B**) Plot of all quantified data involving DNA strand break measurements, showing the timing and magnitude of differences in DNA strand breaks between SCI and Sham groups. Calculated fold change of SCI versus Sham per time point is visualized. hpi, hours post-injury; dpi, days post-injury.

**Figure 3 antioxidants-11-01728-f003:**
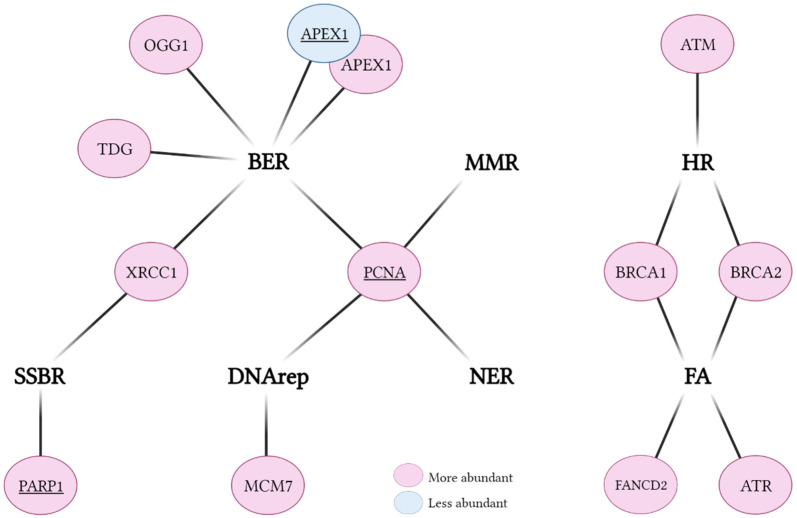
Gene expression alterations of specific DNA repair factors following SCI. DNA repair pathways are indicated in bold. Factors (see Table 3 for further details) validated by independent groups are underlined. BER, base excision repair; MMR, mismatch repair; DNArep, DNA replication; NER, nucleotide excision repair; SSBR, single-strand break repair; HR, homologous recombination; FA, Fanconi anemia.

**Figure 4 antioxidants-11-01728-f004:**
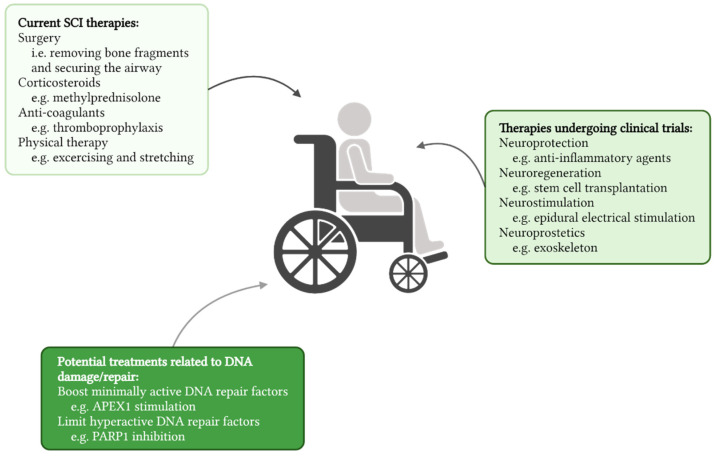
Therapeutic interventions following SCI. In the light box (upper left) currently applied SCI therapies are shown, the medium light box (right) indicates therapies currently undergoing clinical trials, and the dark box (bottom) shows potential DNA damage and repair-related therapeutic avenues that can be further explored.

**Figure 5 antioxidants-11-01728-f005:**
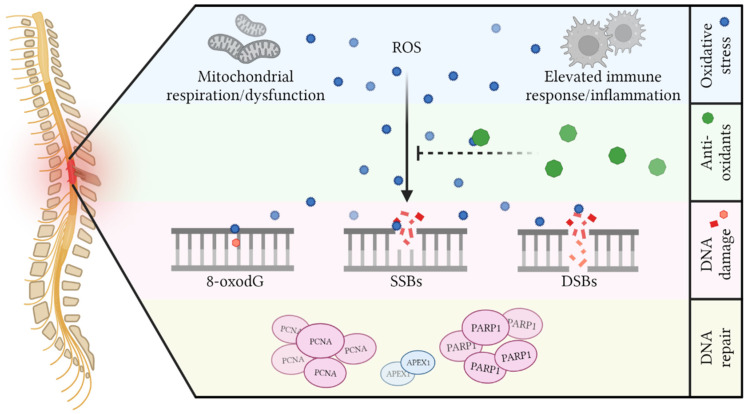
Oxidative DNA damage after SCI results from massive oxidative stress with limited antioxidative capacities. Mitochondrial respiration and ensuing dysfunction, together with elevated immune responses and excessive inflammation, are thought to be the primary producers of ROS in the spinal cord following injury. Due to the intrinsic limited presence and injury-induced loss of antioxidants, ROS can attack the DNA of cells, giving rise to 8-oxodG, single-strand breaks (SSBs), and double-strand breaks (DSBs). Evidence indicates that DNA damage response factors such as PCNA and PARP1 are upregulated, whereas APEX1 is downregulated.

**Table 1 antioxidants-11-01728-t001:** Overview of studies examining oxidative DNA damage following SCI.

	Study	Animal Model	SCI Model	Analysis Method	Time Post-Injury	Effect Size (SCI-Sham) ^1^	Measurement Unit	Fold Change (SCI/Sham) ^1^
	Leski, 2001 [50]	Rat	T13 weight-drop 50–75 g·cm	HPLC with ECD	1 hpi 3 hpi 6 hpi 12 hpi 1 dpi 2 dpi	0.5 1.1 * 0.9 * 0.1 0.1 0.1	8-oxodG/2-dG × 10^−4^	1.4 1.8 * 1.6 * 1.1 1.1 1.1
	Sakurai, 2003 [46]	Rabbit	Transient spinal cord ischemia	8-oxodG IHC	8 hpi 1 dpi 2 dpi	No # Up # Up #	8-oxodG immunoreactivity	
	Bao, 2004 [47]	Rat	T4 extradural compression via aneurysm clip for 60 s	8-oxodG IHC	1 dpi	24 *	8-oxodG+ cells/0.4 mm^2^	30 *
	Takahashi, 2004 [37]	Rabbit	Transient spinal cord ischemia	8-oxodG IHC	8 hpi 1 dpi 2 dpi	No # Up # Up #	8-oxodG immunoreactivity	
	Martin, 2005 [36]	Rat	Sciatic nerve avulsion	8-oxodG IHC	2 dpi 4 dpi 5 dpi 7 dpi 10 dpi	59 * 129 * 357 * 292 * 194 *	8-oxodG+ motor neurons	2.6 * 3.5 * 5.5 * 6.9 * 4.0 *
	Xu, 2005 [29]	Mouse	T12-L3 compression 20 g for 5 min	8-oxodG IHC	1 hpi 6 hpi 12 hpi 1 dpi	Up # Up # Up # Up #	8-oxodG immunoreactivity	
	King, 2006 [48]	Rat	T7-9 hemi-section	8-oxodG IHC	7 dpi	15 † (dorsal) 10 † (ventral)	8-oxodG+ cells/mm^2^	
	Huang, 2007 [10]	Rat	T12 compression 50 g for 5 min	8-oxodG IHC	3 dpi 7 dpi	10 † 4 †	8-oxodG+ cells	
	Kotipatruni, 2011 [49]	Rat	T10 weight-drop 10 g at 1.25 cm	8-oxodG WB	1 dpi 3 dpi 7 dpi 14 dpi 21 dpi	0.36 * 0.44 * 0.57 * 0.41 * 0.32 *	AU	2.5 * 2.8 * 3.4 * 2.7 * 2.3 *
	Sakarcan, 2017 [34]	Rat	T7-10 weight-drop 10 g at 10 cm	8-oxodG ELISA	7 dpi	4.5 *	ng/mg DNA	2.1 *

Color bars correlate with data shown in Figure 1B. ^1^ Estimates of effect size and fold change extracted from graphs in papers. * Statistical difference between SCI and Sham. # No statistics performed. † Not versus Sham (SCI count only). T13, thoracic spine level 13; L3, lumbar spine level 3; HPLC, high-pressure liquid chromatography; ECD, electrochemical detection; IHC, immunohistochemistry; WB, Western blot; hpi, hours post-injury; dpi, days post-injury; AU, arbitrary units.

**Table 2 antioxidants-11-01728-t002:** Overview of studies examining DNA strand breaks following SCI.

	Study	Animal Model	SCI Model	Analysis Method	Time Post-Injury	Effect Size (SCI-Sham) ^1^	Measurement Unit	Fold Change (SCI/Sham) ^1^
	Liu, 2001 [51]	Rat	Sciatic nerve avulsion	Comet assay SSBs	5 dpi 7 dpi 10 dpi	26 † 36 † 23 †	% Comet	
	Martin, 2002 [52]	Rat	Sciatic nerve avulsion	Comet assay SSBs	5 dpi 7 dpi 10 dpi 14 dpi 20 dpi 28 dpi	24 # 35 # 23 # 0 # 0 # 0 #	% Comet	13 # 35 # 23 # 0 # 0 # 0 #
	Genovese, 2005 [56]	Mouse	T6-7 dural compression via 24 g aneurysm clip	PAR IHC	1 dpi	Up #	PAR immunoreactivity	
	Dagci, 2009a [53]				7 dpi 28 dpi	8 # 17 #	Tail%	3.0 # 4.4 #
		Rat	T8-9 micro-scissor cuts	Comet assay DNA damage	7 dpi 28 dpi	16 # 30 #	Tail length	4.2 # 7.0 #
					7 dpi 28 dpi	5 # 7 #	Tail moment	3.5 # 3.3 #
	Dagci, 2009b [54]				7 dpi 28 dpi	10.2 * 26.7 *	Tail%	2.0 * 4.5 *
		Rat	T8-9 micro-scissor cuts	Comet assay DNA damage	7 dpi 28 dpi	6.9 * 10.8 *	Tail length	1.9 * 2.6 *
					7 dpi 28 dpi	3.7 * 10.5 *	Tail moment	3.7 * 9.9 *
	Paterniti, 2010 [57]	Mouse	T5-8 extradural compression via 24 g aneurysm clip	PAR IHC	1 dpi	8 *	% of total tissue area	3 *
	Kotipatruni, 2011 [49]	Rat	T10 weight-drop 10 g at 1.25 cm	Comet assay Strand breaks	1 dpi 7 dpi 21 dpi	75 90 114 *	Comets/section	13 15 19 *
	Ozgonul, 2012 [55]				7 dpi 28 dpi	1.7 * 3.5 *	Tail%	1.7 * 2.3 *
		Rat	T8-9 micro-scissor cuts	Comet assay DNA damage	7 dpi 28 dpi	12.3 * 14.3 *	Tail length	3.5 * 2.8 *
					7 dpi 28 dpi	1.3 * 1.9 *	Tail moment	1.9 * 2.1 *
	Tuxworth, 2019 [58]	Rat	T8 dorsal column crush injury	γ-H2AX IHC	28 dpi	5.2 †	γ-H2AX^+^ pixels/cell	

Color bars correlate with data shown in Figure 1B. ^1^ Estimates of effect size and fold change extracted from graphs in papers. * Statistical difference between SCI and Sham. # No statistics performed. † Not versus Sham (SCI count only). T13, thoracic spine level 13; L3, lumbar spine level 3; HPLC, high-pressure liquid chromatography; ECD, electrochemical detection; IHC, immunohistochemistry; WB, Western blot; hpi, hours post-injury; dpi, days post-injury; AU, arbitrary units.

**Table 3 antioxidants-11-01728-t003:** Overview of studies assessing DNA repair factors following SCI.

Factor	Function	Study	Animal Model	SCI Model	Analysis Method	Cell Type	Effect
**BER**							
OGG1	DNA glycosylase	Kotipatruni, 2011 [49]	Rat	T10 weight-drop 10 g at 1.25 cm	qPCR, WB, and IHC	Total SC	Up
TDG	DNA glycosylase	Kotipatruni, 2011 [49]	Rat	T10 weight-drop 10 g at 1.25 cm	qPCR, WB, and IHC	Total SC	Up
APEX1	AP endonuclease	Sakurai, 2003 [46]	Rabbit	Transient spinal cord ischemia	WB	Neuron	Down
		Bao, 2004 [47]	Rat	T4 extradural compression via aneurysm clip for 60 s	WB and IHC	Total SC	Down
		Dagci, 2009b [54]	Rat	T8-9 micro-scissor cuts	qPCR	Total SC	Down
		Kotipatruni, 2011 [49]	Rat	T10 weight-drop 10 g at 1.25 cm	qPCR, WB, and IHC	Total SC	Up
**SSBR**							
XRCC1 *	Scaffold, Ligase 3 accessory factor	Kotipatruni, 2011 [49]	Rat	T10 weight-drop 10 g at 1.25 cm	qPCR, WB, and IHC	Total SC	Up
PARP1	Strand break response PAR polymerase	Kotipatruni, 2011 [49]	Rat	T10 weight-drop 10 g at 1.25 cm	qPCR, WB, and IHC	Total SC	Up
		Meng, 2015 [68]	Rat	T10 weight drop (not further specified)	qPCR, WB, IHC	Total SC	Up
		Muthaiah, 2019 [78]	Rat	T10-11 weight-drop 10 g at 2.5 cm	qPCR and WB	Total SC	Up
**HR/FA**							
BRCA1 (FANCS)	Accessory factor for transcription and recombination, E3 ubiquitin ligase	Noristani, 2017 [79]	Mouse	T9 full transection or hemi-section	RNA-seq	Microglia	Up
BRCA2 (FANCD1)	Cooperation with RAD51 in recombinational repair	Kotipatruni, 2011 [49]	Rat	T10 weight-drop 10 g at 1.25 cm	qPCR and WB	Total SC	Up
FANCD2	Target for mono-ubiquitination	Noristani, 2017 [79]	Mouse	T9 full transection or hemi-section	RNA-seq	Microglia	Up
ATR	DNA damage sensor kinase	Kotipatruni, 2011 [49]	Rat	T10 weight-drop 10 g at 1.25 cm	qPCR and WB	Total SC	Up
**DSBR**							
ATM	DSB sensor kinase	Kotipatruni, 2011 [49]	Rat	T10 weight-drop 10 g at 1.25 cm	qPCR and WB	Total SC	Up
**DNArep**							
MCM7	Genome replication factor	Chen, 2013 [81]	Rat	T9 weight-drop 10 g at 10 cm	WB and IHC	Total SC	Up
PCNA *	Sliding clamp for polymerase delta and epsilon	Giovanni, 2003 [80]	Rat	T8-9 weight-drop 10 g at 1.75 cm	qPCR, WB, and IHC	Total SC	Up
		Chen, 2013 [81]	Rat	T9 weight-drop 10 g at 10 cm	WB and IHC	Total SC	Up
		Chen, 2016 [82]	Rat	T9 weight-drop 10 g at 10 cm	WB and IHC	Total SC	Up

* Factor also associated with other DNA repair pathways. BER, base excision repair; SSBR, single-strand break repair; HR, homologous recombination; FA, Fanconi anemia; DSBR, double-strand break repair; DNArep, DNA replication; T10, thoracic spine level 10; qPCR, quantitative PCR; WB, Western blot; IHC, immunohistochemistry; RNA-seq, RNA sequencing; SC, spinal cord.

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
