# Peer review of "Oxidative DNA Damage in the Pathophysiology of Spinal Cord Injury: Seems Obvious, but Where Is the Evidence?"

_antioxidants, 2022, doi:10.3390/antiox11091728_

Round 1
Reviewer 1 Report
Manuscript ID: antioxidants-1835751
Manuscript title: Oxidative DNA Damage in the Pathophysiology of Spinal Cord Injury: Seems Obvious, but Where is the Evidence?
Authors thoroughly described the oxidative DNA damage in the spinal cord after ischemia and this review gives a nice information about the changes of oxidative damage in the spinal cord. I have only few comments.
Authors should demonstrate the schematic drawing how to form the 8-oxodG in neurons more clearly although they showed this in Figure 4.
In table 1, authors showed effect size in numeric scale and showed this with graph in Figure 1B. I think it is more understandable to demonstrate the table with figure 1B (especially overlapped study). In addition, others and figure 1A is better to be combined.
Reviewer 2 Report
In this manuscript, Scheijen et al. review the role of oxidative DNA damage in the pathogenic process of spinal cord injury, mainly focused on oxidative DNA damage
Comments and suggestions:
1) In the antioxidant section, the role of NFR2 should be mentioned.
2) It would be interesting to mention the role of lipid peroxidation and iron in oxidative stress after SCI.
3) An illustration of potential pharmacological interventions in SCI would be very informative.
4) It would be interesting to mention the different types of cell death after DNA damage.
Round 2
Reviewer 2 Report
The authors have addressed all of my concerns